# VLAW: Iterative Co-Improvement of Vision-Language-Action Policy and World Model

**Yanjiang Guo** [1 2 *]  **Tony Lee** [1 *]  **Lucy Xiaoyang Shi** [1 *]  **Jianyu Chen** [2]  **Percy Liang** [1]  **Chelsea Finn** [1]

[1]Stanford University, [2]Tsinghua University

## Abstract

The goal of this paper is to improve the performance and reliability of vision-language-action (VLA) models through iterative online interaction. Since collecting policy rollouts in the real world is expensive, we investigate whether a learned simulator—specifically, an action-conditioned video generation model—can be used to generate additional rollout data. Unfortunately, existing world models lack the physical fidelity necessary for policy improvement: they are predominantly trained on demonstration datasets that lack coverage of many different physical interactions (particularly failure cases) and struggle to accurately model small yet critical physical details in contact-rich object manipulation. We propose a simple iterative improvement algorithm that uses real-world roll-out data to improve the fidelity of the world model, which can then, in turn, be used to generate supplemental synthetic data for improving the VLA model. In our experiments on a real robot, we use this approach to improve the performance of a state-of-the-art VLA model on multiple downstream tasks. We achieve a 39.2% absolute success rate improvement over the base policy and 11.6% improvement from training with the generated synthetic rollouts. Videos can be found at this anonymous website: https://sites.google.com/view/vlaw-arxiv.

## 1. Introduction

Vision-language-action (VLA) models have achieved great success in robot manipulation by training on large-scale demonstration data (Intelligence et al., 2025b; Kim et al., 2024; Shi et al., 2025; Guo et al., 2025b; Zhang et al., 2024;

*Core contributors [1]Stanford University [2]Tsinghua University. Correspondence to: Yanjiang Guo <yjguo@stanford.edu>.

*Proceedings of the 43rd International Conference on Machine Learning*, Seoul, South Korea. PMLR 306, 2026. Copyright 2026 by the author(s).

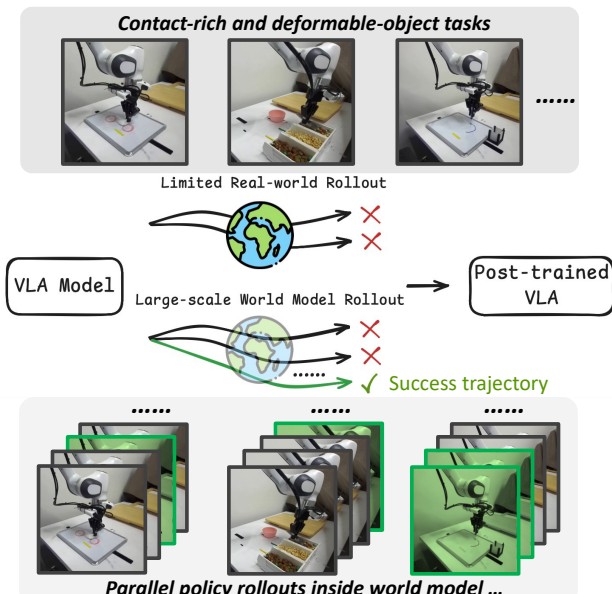

*Figure 1.* VLA model roll-outs in the real world are time-consuming and unscalable. In VLAW, we first learn an action-conditioned world model using limited real-world online rollouts, which in turn generates large-scale synthetic data in imagination.

Chen et al., 2025). Recent studies further show that VLA models can benefit substantially from post-training on online interaction rollous (Intelligence et al., 2025a). However, in real-world robotic settings, collecting online policy roll-out trajectories requires significant human labor, such as resetting the environment and monitoring robot execution, which is expensive and time-consuming (Atreya et al., 2025; Jain et al., 2025). As a result, the number of online rollouts available for VLA models is often limited, restricting the effectiveness and scalability of post-training.

Instead of relying solely on real-world policy rollouts, learning an action-conditioned world model to generate synthetic rollouts in imagination offers a promising alternative (Team et al., 2025; Li et al., 2024; Team, 2025b). However, we find that existing world models lack the physical fidelity required for effective policy improvement. As noted in prior works, these models tend to be overly optimistic about predicted trajectories, as they are trained predominantly on

demonstration datasets that lack coverage of diverse physical interactions, especially failure cases (Quevedo et al., 2025). Moreover, they struggle to accurately model small yet critical physical details in contact-rich manipulation and can produce blurry visual predictions (Guo et al., 2025a). Consequently, existing action-conditioned world models have largely focused on relatively simple pick-and-place motions and often fail to generate reliable synthetic data for complex tasks involving frequent collisions or deformable objects.

In this paper, we propose a simple yet scalable framework, VLAW, that iteratively improves VLA models via world-model rollouts, as shown in Figure 2. We first learn a physically-grounded world model by finetuning on online rollout data, which includes many failure cases. We find that after training on online rollout data, the world model learns to capture the complex dynamics encountered during policy execution, substantially improving its ability to model both success and failure cases. The improved world model is subsequently used to generate large-scale, high-fidelity synthetic trajectories, which are automatically annotated using a vision–language reward model (Lee et al., 2026). During policy optimization, we only use stable supervised learning objectives that can easily scale to large expressive models (e.g., flow-matching policies with intractable action probabilities (Intelligence et al., 2025b)), as opposed to dynamic programming/bootstrapping or policy gradients.

The core contribution of this paper is a simple and scalable world-model-based reinforcement learning framework for improving state-of-the-art VLA policies in the real world. In our experiments, we use the widely used real-robot platform DROID (Khazatsky et al., 2024). We start from a pretrained VLA policy, $\pi_{0.5}$ (Intelligence et al., 2025b) and an action-conditioned world model, Ctrl-World (Guo et al., 2025a). We first verify that, using policy online rollout data, we learn a physically grounded generative world model that can accurately model both success and failure trajectories, which is essential for generating useful synthetic data. In addition, to obtain a reward model for robot tasks, we fine-tune Qwen3-VL (Team, 2025a; Lee et al., 2026) on real-robot rollout data. Finally, using the synthetic data generated by the world model, we improve the pretrained $\pi_{0.5}$ across many downstream contact-rich manipulation tasks that involve deformable objects in a multi-task setup, outperforming baseline with 11.6%.

## 2. Related Works

### 2.1. Post-training Vision-Language-Action Models

Vision–language–action (VLA) models have achieved remarkable success in robotic manipulation tasks (Intelligence et al., 2025b; Pertsch et al., 2025; Liu et al., 2025a; Cui

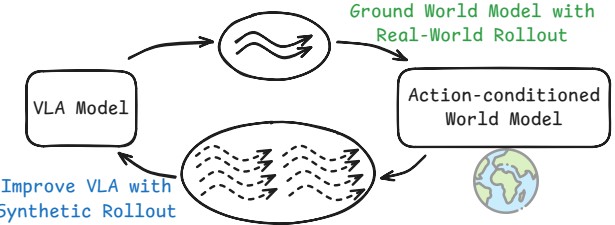

*Figure 2.* Policy online rollout data can help **ground** the pretrained world model in downstream tasks. Once the world model is grounded, we can generate massive data for policy learning.

et al., 2025; Hu et al., 2024; Guo et al., 2024; Zhang et al., 2026). A common approach is to train the VLA on large-scale data and then perform supervised fine-tuning on target tasks (Zhang et al., 2025a; Black et al., 2024; Zhang et al., 2025b). Beyond supervised fine-tuning, improving VLA policies using online rollout data has emerged as a promising direction (Intelligence et al., 2025a; Guo et al., 2025b; Lu et al., 2025; Zang et al., 2025; Huang et al., 2024; Cheng et al., 2025). Some prior works adopt on-policy reinforcement learning methods, such as PPO (Schulman et al., 2017) or GRPO (Shao et al., 2024), to improve VLA policies.

However, standard on-policy reinforcement learning typically requires a large number of rollouts and is therefore primarily validated in simulation environments (Li et al., 2025b;a; Liu et al., 2025b). Moreover, state-of-the-art VLA models are often trained with flow-matching objectives, which do not provide explicit policy likelihoods, making conventional policy-gradient methods difficult to apply. To enable policy learning in real-world settings, $\pi_{0.6}^*$ (Intelligence et al., 2025a) instead adopts an offline or batch reinforcement learning formulation with an advantage-conditioned supervised learning objective. Similarly, in our setting, we perform iterative policy improvement using batches of real-world rollout data together with world-model–generated synthetic data, and update the policy exclusively through stable supervised fine-tuning objectives.

### 2.2. World Models for Decision Making

Action-conditioned world models predict future outcomes given current observations and actions, and are also referred to as forward dynamics models. Many works leverage such models for model-based reinforcement learning (Hafner et al., 2020; Hansen et al., 2022; Oh et al., 2015; Wu et al., 2024) and visual planning (Finn & Levine, 2017; Ebert et al., 2018; Xie et al., 2019; Dasari et al., 2019; Yang et al., 2023). Among these, the most closely related approaches to ours are DayDreamer (Wu et al., 2023), SOLAR (Zhang et al., 2019) and World4rl (Jiang et al., 2025), which also operate in real-world visual model-based reinforcement learning settings. However, due to limited model capacity and data scale, these earlier methods often learned task-specific dynamics models.

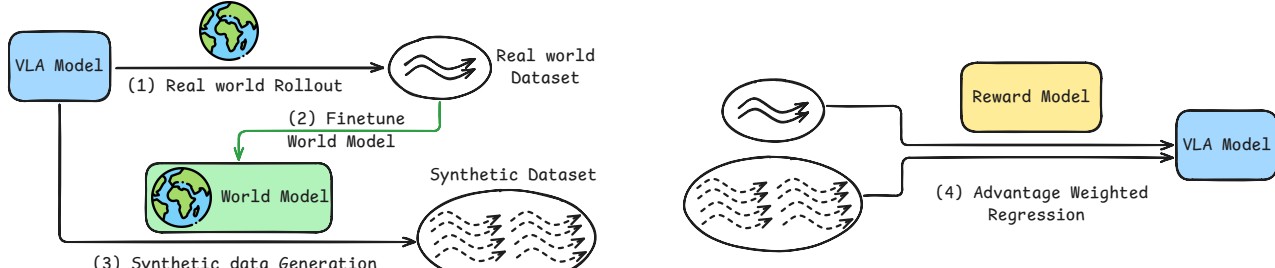

1. Learning accurate world model

2. Regularized RL with Expressive Policy

*Figure 3.* Detailed pipeline for VLAW: (1) We first roll out the policy in the real world to collect a small set of online trajectories. (2) We then fine-tune a pretrained action-conditioned world model on these policy rollout data, grounding the world model in the target tasks and improving its predictive fidelity. (3) Using the resulting world model, we generate large-scale synthetic trajectories through closed-loop interactions between the policy and the world model. (4) Finally, we optimize the VLA policy using both real-world and synthetic data, with reward automatically assessed by a vision–language reward model.

With recent advances in video diffusion models (Ren et al., 2025; Ball et al., 2025; Mei et al., 2026), it has become feasible to train multi-task action-conditioned world models that can generate realistic future visual observations (Chen et al., 2024; Gao et al., 2025; Zhu et al., 2024; 2025; Sharma et al., 2026). Despite this progress, accurately modeling complex physical dynamics remains a fundamental challenge, as widely observed in prior world-model literature (Guo et al., 2025a), likely because these models are trained on offline robotics datasets usually consisting primarily of demonstrations. To address this challenge, we leverage online policy rollout data to ground a pretrained world model in new environments, thereby improving its accuracy around the policy's state–action distribution.

## 3. Preliminaries

**Problem Setting.** We study a multi-task robotic manipulation problem, where each task is specified by a language instruction $I$ and is modeled as a Markov decision process (MDP) $\mathcal{M}_I = (\mathcal{S}, \mathcal{A}, P, R_I, \gamma)$. Here, $\mathcal{S}$ denotes the state space, $\mathcal{A}$ the action space, $P(s_{t+1} \mid s_t, a_t)$ the transition dynamics, $R_I$ the task-dependent reward function, and $\gamma$ the discount factor. At the beginning of training, we are given a pretrained vision–language–action (VLA) policy $\pi_\theta$ and an action-conditioned world model $M_\phi$. The policy maps the current state and instruction to an action distribution, $a_t \sim \pi_\theta(\cdot \mid s_t, I)$, while the world model predicts the next state conditioned on the current state and action, $\hat{s}_{t+1} \sim M_\phi(\cdot \mid s_t, a_t)$, where $\hat{s}_{t+1}$ denotes the predicted next state.

The policy is allowed to collect online roll-outs in the real environment, resulting in trajectories $\tau^i_{\text{real}} = \{s_0, a_0, \ldots, a_{T-1}, s_T\}$. Each trajectory is labeled with a task-level reward $r_i$ indicating success or failure. Our goal is to leverage online interaction to iteratively improve the policy so that it performs well across all tasks.

**World Model Generated Trajectories.** In addition to real-world interaction, we can roll out the policy inside the world model. Starting from an initial state $s_0$ sampled from a real trajectory, the policy and world model interact in a closed loop via $a_t \sim \pi_\theta(\cdot \mid \hat{s}_t, I)$ and $\hat{s}_{t+1} \sim M_\phi(\cdot \mid \hat{s}_t, a_t)$. By iterating this process, we auto-regressively generate a complete imagined trajectory $\tau^j_{\text{syn}} = \{s_0, a_0, \hat{s}_1, a_1, \ldots, a_{T-1}, \hat{s}_T\}$.

## 4. Co-Improvement of VLA and World Model

In this section, we describe the details of our method. The overall pipeline consists of the following steps:

1. **World model post-training (Sec. 4.1):** We finetune the world model $M$ using real-world rollout data $\mathcal{D}_{\text{real}}$, jointly training it with the original DROID dataset $\mathcal{D}_{\text{DROID}}$ to maintain broad coverage. In addition, we finetune the vision-language reward model $R$ on $\mathcal{D}_{\text{real}}$ to improve reward accuracy.

2. **VLA policy post-training (Sec. 4.2):** Using the updated world model, we generate a synthetic dataset $\mathcal{D}_{\text{syn}}$ and apply the reward model $R$ to identify successful trajectories, yielding a filtered dataset $\mathcal{D}^+_{\text{syn}}$. This dataset is then used to finetune the VLA policy.

3. We alternate between Steps 1 and 2, iteratively improving both the world model and the policy.

The overall pipeline is summarized in Algorithm 1 and Figure 3. In Sec. 4.3, we provide a detailed analysis showing that our update procedure can be interpreted as an approximation to policy optimization under a regularized reinforcement learning framework.

### 4.1. World Model Learning with Real Roll-outs

**Real World Policy Roll-outs.** Previous work has identified two major challenges in learning effective world models: (1) *over-optimism*, as training data is dominated by successful

demonstrations; and (2) *limited physical fidelity*, particularly when modeling complex dynamics involving frequent contacts or deformable objects.

To address these issues, we get $K$ trajectories by rolling out the policy in the real world, forming a dataset $\mathcal{D}_{\text{real}} = \{\tau_{\text{real}}^1, ..., \tau_{\text{real}}^K\}$, we also assign a sparse reward $r_\tau \in \{0, 1\}$ to each trajectory to indicate success or not every time we reset robot.

**Training Objective.** $\mathcal{D}_{\text{real}}$ captures diverse physical interactions encountered during execution, including both success and failure cases, and is used to finetune a pretrained world model. Specifically, we initialize from the pretrained Ctrl-World model (Guo et al., 2025a), a strong diffusion-based world model trained on the full DROID dataset $\mathcal{D}_{\text{DROID}}$. Finetuning on the online rollout dataset $\mathcal{D}_{\text{real}}$ follows the original diffusion objective (Blattmann et al., 2023):

$$\mathcal{L}_{\mathcal{D}_{\text{real}}} = \mathbb{E}_{x_0, \epsilon, t'} \|\hat{x}_0(x_{t'}, t', c) - x_0\|^2, \qquad (1)$$

where the prediction target $x_0 = o_{t+1:t+H}$ is sampled from $\mathcal{D}_{\text{real}}$, $x_{t'} = \sqrt{\bar{\alpha}_{t'}} x_0 + \sqrt{1 - \bar{\alpha}_{t'}} \epsilon_{t'}$ denotes the noised future at diffusion step $t' \in [0, T']$ under the noise schedule $\bar{\alpha}_{t'}$, and $c$ represents all conditioning inputs, including the action chunk $a_{t:t+H}$ and the current observation $o_t$.

**Progressively Growing Dataset and Co-training.** During successive iterations, we continuously append newly collected real-world trajectories into the dataset: $\mathcal{D}_{\text{real}} = \mathcal{D}_{\text{real}} \cup \tau_{\text{real}}^i$. To prevent overfitting to the limited online rollout data, we also co-train with the original DROID dataset $\mathcal{D}_{\text{DROID}}$ for regularization. The final training objective is:

$$\mathcal{L} = \mathcal{L}_{\mathcal{D}_{\text{real}}} + \lambda \mathcal{L}_{\mathcal{D}_{\text{DROID}}} \qquad (2)$$

where $\lambda$ controls the strength of the regularization.

**Finetuning Reward Model.** To keep our pipeline simple and scalable, we leverage a general-purpose vision-language model, Qwen3-VL-4B-Instruct (Team, 2025a; Lee et al., 2026), to assess whether a trajectory succeeds or not. However, we find that the zero-shot VLM is not accurate enough, so in the first iteration, we fine-tune the VLM with the success labels $r_\tau$ in $\mathcal{D}_{\text{real}}$.

In implementation, the reward model takes as input a trajectory video $\tau_{\text{real}}^i$ together with a query asking whether the task instruction $I^i$ is successfully completed. We classify a trajectory as successful if the probability assigned to the 'yes' token exceeds a threshold $\alpha$. By adjusting $\alpha$, we can make the reward model more or less conservative.

$$R(\tau^i) = \mathbf{1}\big[P(\text{`yes'} \mid \tau^i, I^i) > \alpha\big], \qquad (3)$$

### 4.2. Iterative Improvement for VLA Policy

**Scalable Training Pipeline.** Once we have a good learned world model and reward model, then we can use it to cheaply

---

**Algorithm 1** VLAW

**Require:** Pretrained VLA policy $\pi_\theta$; pretrained world model $M_\phi$; reward model $R$; real-world rollout budget $K$; synthetic rollout budget $N$; iterations $K_{\text{iter}}$; reward threshold $\alpha$

**Output:** Post-trained policy $\pi_\theta$ and world model $M_\phi$

1: Initialize real-world dataset $\mathcal{D}_{\text{real}} \leftarrow \emptyset$
2: **for** $i = 1$ to $K_{\text{iter}}$ **do**
3:    **(1) Real-world rollouts**
4:    Roll out $\pi_\theta$ in the real world to collect $\tau_{\text{real}}^1, \ldots, \tau_{\text{real}}^K$
5:    Append collected trajectories to $\mathcal{D}_{\text{real}}$, success trajectories in $\mathcal{D}_{\text{real}}^+$
6:    **(2) World model and reward model post-training**
7:    Update $M_\phi$ using $\mathcal{D}_{\text{real}}$ and $\mathcal{D}_{\text{DROID}}$ according to Eq. (1) and Eq. (2)
8:    **(3) Synthetic rollout generation with reward label**
9:    Roll out $\pi_\theta$ in $M_\phi$ to generate $\mathcal{D}_{\text{syn}} = \tau_{\text{syn}}^1, \ldots, \tau_{\text{syn}}^N$
10:   Apply reward model $R$ with threshold $\alpha$ (Eq. (3)) to obtain $\mathcal{D}_{\text{syn}}^+$
11:   **(4) Policy post-training**
12:   Update $\pi_\theta$ on $\mathcal{D}_{\text{real}}^+ \cup \mathcal{D}_{\text{syn}}^+$ using the flow-matching objective in Eq. (4)
13: **end for**
14: **return** $\pi_\theta$, $M_\phi$

---

generate a large amount of synthetic data. In principle, many different algorithms could be used to leverage this data, including a variety of sophisticated reinforcement learning methods. Because we want to easily scale to large, flow-matching based VLA policies, we choose to use the one of the simplest possible methods for incorporating this synthetic data.

Specifically, we generate $N$ trajectories by rolling out the policy in imagination: $\mathcal{D}_{\text{syn}} = \{\tau_{syn}^1, ..., \tau_{syn}^N\}$. We then apply the finetuned reward model to identify successful trajectories and construct a filtered dataset containing only success cases: $\mathcal{D}_{\text{syn}}^+ = \{\tau_{syn}^{i_1}, ..., \tau_{syn}^{i_n}\}$, where $i_1, ..., i_n$ is the index of success trajectory.

**Policy Learning Objective.** We update the $\pi_{0.5}$ policy using a weighted flow-matching objective over both real-world rollouts and world-model–generated data. After filtering for successful trajectories, we assign a binary weight $w(o, a) = 1$ to transitions from successful trajectories and $w(o, a) = 0$ to transitions from failed trajectories:

$$\begin{aligned} \mathcal{L} &= \mathbb{E}_{(o,a) \sim \mathcal{D}_{\text{syn}} \cup \mathcal{D}_{\text{real}}} w(o, a) \, \mathcal{L}_{\text{FM}}(\theta; o, a) \\ &= \mathbb{E}_{(o,a) \sim \mathcal{D}_{\text{syn}}^+ \cup \mathcal{D}_{\text{real}}^+} \mathcal{L}_{\text{FM}}(\theta; o, a), \end{aligned} \qquad (4)$$

where $\mathcal{L}_{\text{FM}}(\theta; o, a)$ denotes the flow-matching loss for an observation–action pair $(o, a)$.

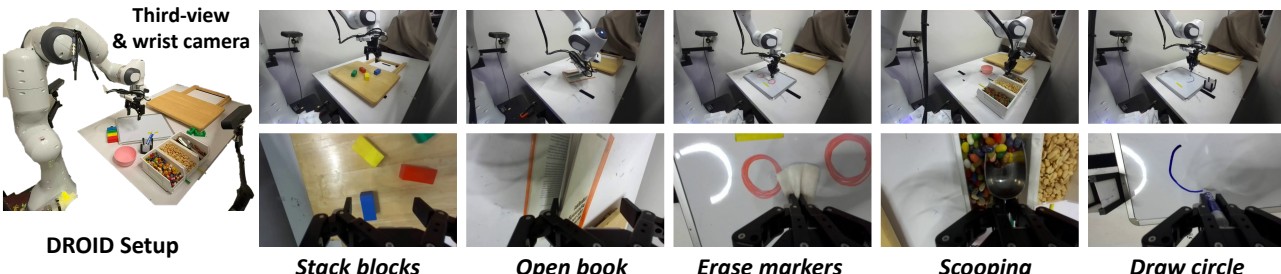

**Figure 4.** Our experiments are conducted on the DROID platform and cover five task categories, as illustrated in the figure. These tasks involve complex physical interactions, including frequent contact and deformable objects, which are challenging to model in traditional simulations.

### 4.3. Relation to Regularized Reinforcement Learning

In this subsection, we show that the policy update in Eq. 4 can be view as policy optimization under a regularized reinforcement learning (RL) framework (Peng et al., 2019) with certain approximations.

Under the regularized RL setting, we constrains the learned policy to remain close to a reference policy $\pi_{\mathrm{ref}}$ while optimizing reward. This yields the following regularized objective:

$$J(\theta) = \mathbb{E}_{\tau \sim \rho_{\pi_\theta}}[R(\tau)] - \beta \, \mathbb{E}_{o \sim \rho_{\pi_\theta}}[D(\pi_\theta(\cdot \mid o) \,\|\, \pi_{\mathrm{ref}}(\cdot \mid o))] \tag{5}$$

where $D(\cdot\|\cdot)$ denotes a KL divergence measure and $\beta > 0$ controls the strength of the regularization. The optimal improved policy admits a closed-form solution given by:

$$\pi^\star(a \mid o) \propto w(o,a)\pi_{\mathrm{ref}}(a \mid o), w(o,a) = \exp\left(\frac{A^{\pi_{\mathrm{ref}}}(o,a)}{\beta}\right)$$

where $\pi_{\mathrm{ref}}$ denotes a reference policy, and $A^{\pi_{\mathrm{ref}}}(o,a)$ is the corresponding advantage function, and $\beta$ is a temperature parameter controlling the strength of the regularization. We can define a surrogate divergence which measures how well $\pi_\theta$ matches samples drawn from $\pi^\star$ under the flow-matching loss:

$$D_{\mathrm{FM}}(\pi^\star(\cdot \mid o), \pi_\theta(\cdot \mid o)) \;\triangleq\; \mathbb{E}_{a \sim \pi^\star(\cdot \mid o)}\big[\mathcal{L}_{\mathrm{FM}}(\theta; o, a)\big], \tag{6}$$

Using this divergence, we can project policy to the optimal solution with :

$$\theta^\star = \arg\min_\theta \; \mathbb{E}_{(o,a) \sim \mathcal{D}}\Big[w(o,a)\,\mathcal{L}_{\mathrm{FM}}(\theta; o, a)\Big], \tag{7}$$

which is the weighted regression objective used in our policy update equation 4. More detailed derivations are provided in Appendix A.

## 5. Experiments

In this section, we conduct extensive experiments on complex real-world tasks involving frequent collisions and de-

formable objects. Our experiments are designed to answer the following questions:

1. Can we learn a high-fidelity action-conditioned world model for contact-rich and deformable-object tasks that accurately models both successful and failed trajectories?
2. Can the synthetic data generated by the world model improve VLA policy performance?
3. Can the policy and world model be continuously improved through an iterative training process in a multi-task setting?

### 5.1. Experimental Settings

**Setups and Tasks.** We conduct experiments on the DROID platform (Khazatsky et al., 2024). In the DROID setup, a Franka Panda arm is equipped with a Robotiq gripper. Observations are captured using two third-person cameras and one wrist-mounted camera, as illustrated in Figure 4. We evaluate our method on five categories of contact-rich tasks, described below. More task details can be found in Appendix B.

- **Stacking:** Four colored blocks are randomly placed on the table at the beginning of each episode. The robot receives the instruction: "stack block $A$ on block $B$," where $A, B \in \{\text{red}, \text{green}, \text{blue}, \text{yellow}\}$.
- **Open Book:** A book is randomly placed on the table at the start of each episode. We evaluate performance across four different books. The robot is instructed to "open the book cover."
- **Erase Marks:** One to three marker drawings are randomly drawn on a whiteboard. The robot receives the instruction: "erase all marks using a tissue."
- **Scooping:** The robot uses a scoop to transfer snacks into a bowl. Both the scoop and the bowl are randomly placed within the workspace. The instruction is: "transfer some $A$ to the bowl," where $A \in \{\text{peanuts}, \text{candies}, \text{almonds}\}$.
- **Drawing:** The robot is instructed to draw a complete circle on a whiteboard using a marker.

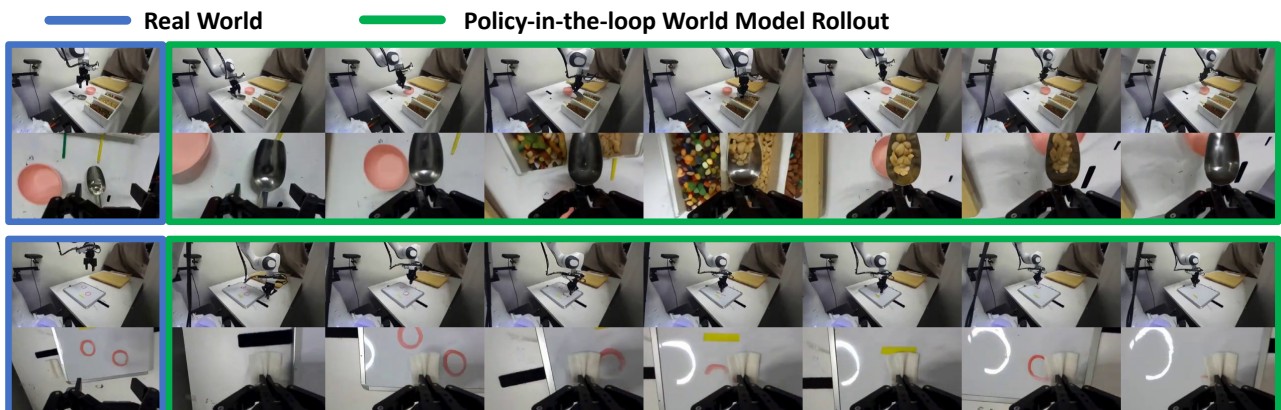

*Figure 5.* Examples of long-horizon policy-in-the-loop rollouts within the world model starting from the initial observation. The policy $\pi_{0.5}$ is rolled out for 20 iterations (20 seconds). The post-trained world model accurately captures contact-rich physical dynamics. Top: scooping peanuts into a new bowl. Bottom: erasing marker drawings with a tissue.

| Method | (1) Video Quality Metrics | | | | | (2) Event Confusion Matrix | | | |
|---|---|---|---|---|---|---|---|---|---|
| | PSNR ↑ | SSIM ↑ | LPIPS ↓ | FID ↓ | FVD ↓ | TP ↑ | FN ↓ | **TN ↑** | **FP ↓** |
| Pretrained Ctrl-world | 16.32 | 0.634 | 0.347 | 41.03 | 225.13 | - | - | - | - |
| Pretrained Ctrl-world + Expert Rollout | 19.87 | 0.748 | 0.189 | 12.76 | 99.98 | 28 | 2 | 9 | 11 |
| Pretrained Ctrl-world + Expert Rollout + Online Rollout | **21.77** | **0.784** | **0.136** | **9.58** | **64.12** | 26 | 4 | 19 | 1 |

*Table 1.* We replay recorded action sequences in the world model. (1) We evaluate video quality metrics on 256 replayed clips, each 5 seconds long. All metrics are computed using the wrist-view camera, as this viewpoint best captures object interactions during manipulation. (2) The interaction phase is the primary source of errors. Therefore, we report an event-level confusion matrix on 50 clips involving physical interactions. For each clip, we label the interaction outcome (success or failure) and compare the model predictions against real-world outcomes.

**Base Models and Hyperparameters.** We use $\pi_{0.5}$ (Intelligence et al., 2025b) as the base vision–language–action (VLA) model and Ctrl-World (Guo et al., 2025a) as the base world model. For each task category, we collect 25 expert demonstrations and finetune $\pi_{0.5}$ on this data to warm-start the policy, which serves as our base policy. The reward model is initialized from Qwen3-VL-4B-Instruct (Team, 2025a).

In each iteration, we roll out 50 trajectories per task category in the real world. We finetune the world model for 50K training steps using these rollout trajectories. We then generate 500 synthetic trajectories per task using the updated world model to form the synthetic dataset. The reward model is additionally finetuned using rollout data from the first iteration to improve reward accuracy. The policy is updated with 2k steps with batch size 256. We perform a total of two iterations of this procedure.

### 5.2. Can we learn an accurate action-conditioned world model for contact-rich tasks?

**Action replay inside the world model.** We evaluate the fidelity of the learned world model and study the contribu-

tion of online rollout data by replaying real-world action sequences inside the world model. Specifically, we randomly select a starting frame from a real-world trajectory and auto-regressively feed a 5-second sequence of recorded action chunks to the world model, starting from the same frame. We compare our post-trained world model against two baselines: the original pretrained world model and a model finetuned only on expert demonstration data.

We use two categories of metrics to quantitatively evaluate video prediction quality:

- **(1) Video distance metrics:** These include pixel-level metrics (PSNR (Hore & Ziou, 2010) and SSIM (Wang et al., 2004)) as well as learned perceptual and distributional metrics (LPIPS (Zhang et al., 2018), FID (Heusel et al., 2017), and FVD (Unterthiner et al., 2018)).
- **(2) Interaction event confusion matrix:** Correctly predicting the outcome of object interactions is the most challenging aspect of action-conditioned world modeling. We filter replayed clips that involve object interactions and classify each interaction as success or failure. We then evaluate whether the predicted outcome aligns with the real-world result.

Quantitative results are reported in Table 1. Finetuning with

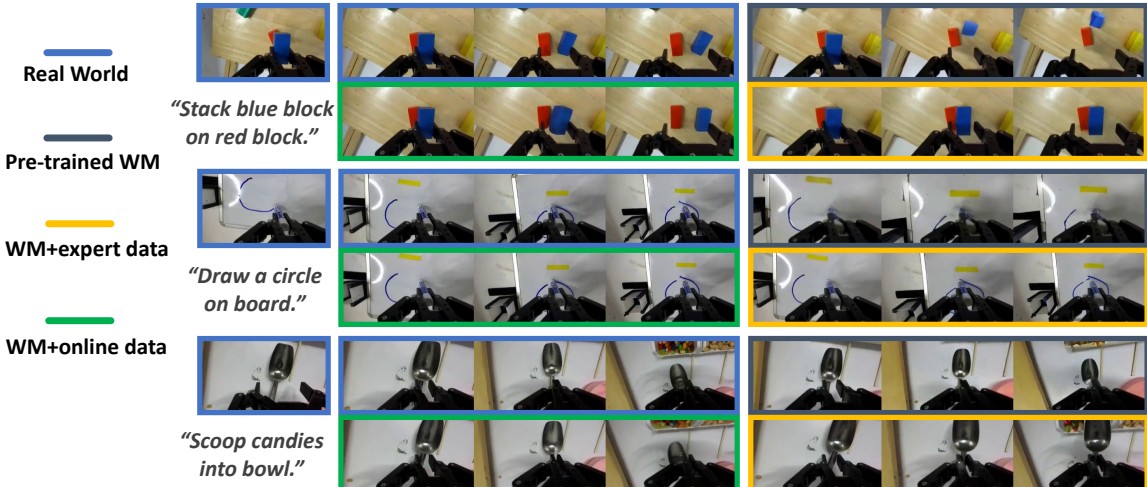

*Figure 6.* Conditioned on the same initial frame and identical action sequences (five chunks), we roll out trajectories inside different world models. The pretrained Ctrl-World model is insufficiently accurate for these contact-rich tasks. World models fine-tuned only on expert trajectories tend to be overly optimistic. In contrast, the world model fine-tuned on policy online rollout data accurately captures the underlying physical dynamics and is well aligned with real-world outcomes. Only the wrist-view camera is shown due to space limitations. Zoom in for better comparisons.

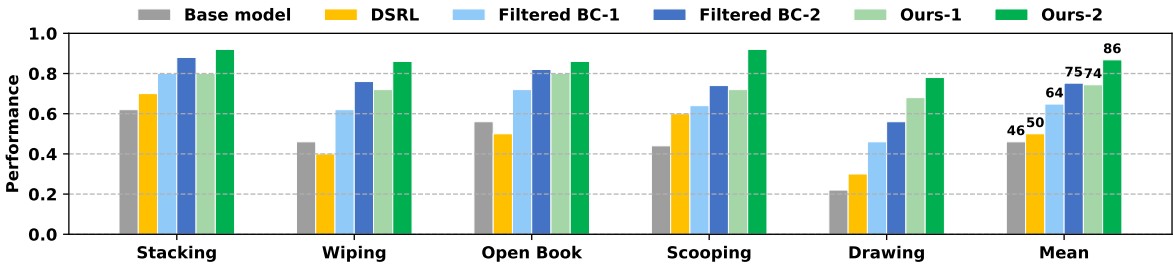

*Figure 7.* Success Rate Improvement Comparison with Baselines. We perform two rounds of iterative training. "Ours-1" denotes the VLAW method after the first round of online rollouts. Overall, VLAW consistently outperforms both the filtered BC and DSRL baselines in the multi-task setting.

online rollout data is crucial for world model performance: all video quality metrics improve substantially compared to both baselines. Moreover, by training on mixed success and failure trajectories, the world model largely eliminates the over-optimistic bias observed when training only on expert demonstrations. In particular, false-positive interaction predictions are significantly reduced. We provide qualitative visualizations of interaction replay in Figure 6.

**Policy-in-the-loop rollout.** We further evaluate the world model by rolling out the policy directly inside the learned model. Although evaluated tasks involve complex, contact-rich interactions, and we find that the post-trained world model maintains high visual fidelity and physical plausibility even for long-horizon rollouts of up to 20 seconds. Example rollouts are shown in Figure 5. This long-horizon stability enables effective search for successful trajectories within the world model, which we subsequently leverage for policy improvement.

### 5.3. Can world model generated data improve VLA policy performance?

**Baselines.** Our goal is to leverage real-world online interaction data to improve the VLA policy while minimizing physical rollouts. Under this setting, we compare our method against two baselines that do not utilize a world model:

- **(1) Filtered BC**, which filters successful trajectories from real-world rollouts and performs supervised fine-tuning on these trajectories. We control the real world rollout number the same as our method for fair comparison (50 rollouts for each category of tasks).
- **(2) DSRL** (Wagenmaker et al., 2025), which improves the $\pi_{0.5}$ policy by optimizing its noise space through online exploration, we control the online rollout number the same as other methods.

**Large-scale rollout visualizations.** We visualize parallel rollouts generated by the world model in Figure 8. Starting from an initial frame recorded in the real world (GT), we search for successful trajectories entirely within the world

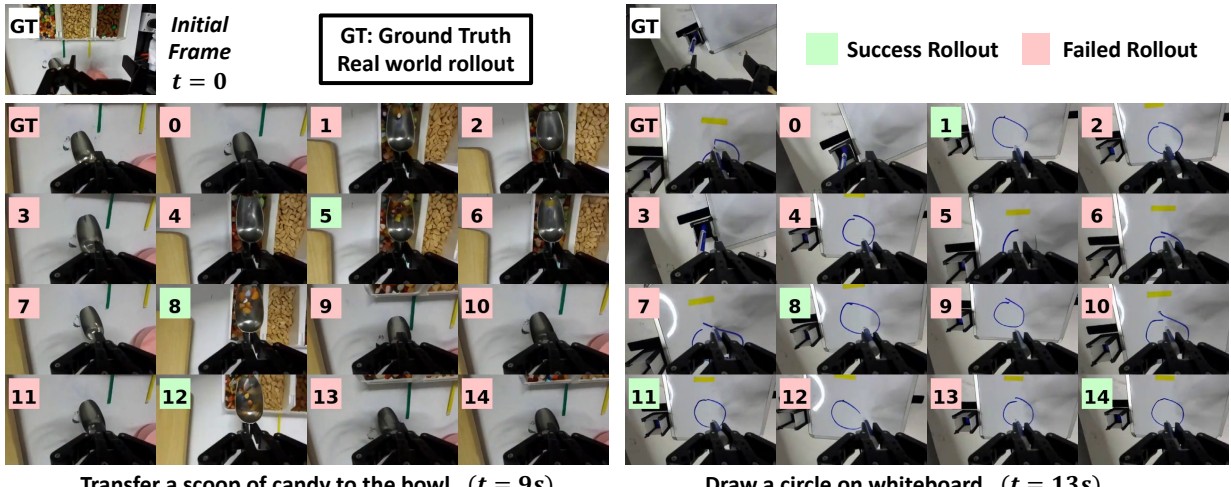

*Figure 8.* GT denotes the real-world rollout, while $0 \sim 14$ denotes diverse trajectories imagined by the world model, all rollouts from the same GT initial frame with $\pi_{0.5}$. In the real-world rollout, the robot fails to grasp the scoop (left, GT) and fails to draw a complete circle (right, GT). With the help of a world model, we can **search successful trajectories for failure cases**, which can be useful for policy learning.

model. These successful imagined trajectories provide additional supervision for policy learning, enabling the policy to progressively overcome failure cases and improve task performance.

**Reward model analysis.** We use a learned reward model to filter successful trajectories from world model–generated rollouts. As described in the method section, a trajectory is considered successful only if the probability assigned to the 'yes' token exceeds a predefined threshold. This thresholding strategy substantially reduces false-positive trajectories. Additional details and analyses of the reward model are provided in Appendix C.

**Results.** The success rate improvements are shown in Figure 7. DSRL achieves limited gains in our multi-task setting. We hypothesize that this is because reinforcement learning becomes significantly harder to optimize across diverse tasks, and because DSRL constrains optimization to the noise space of the $\pi_{0.5}$ policy rather than updating the model parameters directly, which limits the expressive capacity of the policy. Filtered BC improves performance over two iterations by leveraging successful real-world trajectories. In contrast, by generating large-scale synthetic rollouts and selectively filtering successful trajectories, VLAW achieves substantially larger performance gains across all tasks.

**Ablations.** We conduct ablation studies on (1) the number of world model rollouts and (2) whether real-world rollout data is included during policy finetuning. We evaluate these ablations on the most challenging drawing task, with results shown in Figure 9. Reducing the amount of synthetic rollout data leads to noticeable performance degradation, and removing real-world success trajectories during finetuning further harms performance, highlighting the importance of

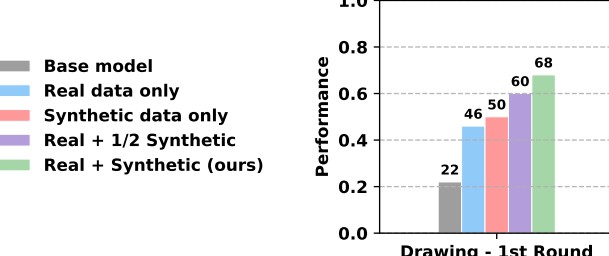

*Figure 9.* We conduct ablation studies on (1) the amount of synthetic data used for policy fine-tuning (reducing from 500 to 250 trajectories) and (2) whether real-world rollout data (50 trajectories) is included during fine-tuning. We observe that either decreasing the number of synthetic trajectories or removing the real-world dataset leads to a performance degradation.

both components.

## 6. Conclusions and discussions

In this paper, we propose VLAW, an iterative improvement pipeline that jointly enhances both the vision–language–action (VLA) policy and the action-conditioned world model. We demonstrate that VLAW consistently improves performance across multiple contact-rich manipulation tasks. Although the learned world model achieves high fidelity on the downstream tasks from which online data are collected, our current evaluation is limited to five task categories. Scaling online rollout data to a broader and more diverse set of tasks is a promising direction for future work. We believe that, as base video models continue to advance and large-scale robot interaction data become increasingly available, world-model-based training will provide a powerful new paradigm for learning generalist robotic policies.

## Impact Statement

This paper presents work whose goal is to advance the field of Machine Learning. There are many potential societal consequences of our work, none which we feel must be specifically highlighted here.

## Acknowledgment

This work was supported by The Robotics and AI Institute and ONR grant N00014-22-1-2621.

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

# A. Relation to Regularized Reinforcement Learning.

In this part, we relate the policy update in Eq. 4 to policy optimization under a regularized reinforcement learning (RL) framework with certain approximations. Our VLA policy is trained with a flow-matching objective and does not provide a tractable action log-likelihood, so standard KL-based derivations do not apply directly. Under the regularized RL setting, the optimal improved policy admits a closed-form solution given by:

$$\pi^\star(a \mid o) \; \propto \; \pi_{\text{ref}}(a \mid o) \, \exp\!\left(\frac{A^{\pi_{\text{ref}}}(o, a)}{\beta}\right), \tag{8}$$

where $\pi_{\text{ref}}$ denotes a reference policy, $A^{\pi_{\text{ref}}}(o, a)$ is the corresponding advantage function, and $\beta$ is a temperature parameter controlling the strength of the regularization.

Since the target distribution $\pi^\star$ is generally not representable within a finite parametric policy class, policy improvement is typically performed via a *projection step*, which fits a parametric policy $\pi_\theta$ to $\pi^\star$ by minimizing a divergence $D$:

$$\theta^\star = \arg\min_\theta \; \mathbb{E}_{o \sim \mathcal{D}}\Big[D(\pi^\star(\cdot \mid o), \pi_\theta(\cdot \mid o))\Big]. \tag{9}$$

**AWR for flow-matching policies.** In standard Advantage-Weighted Regression (AWR) (Peng et al., 2019), the divergence $D$ is chosen to be the KL divergence, which results in a weighted log-likelihood objective. However, because our VLA policy is trained using a flow-matching objective $\mathcal{L}_{\text{FM}}(\theta; o, a)$ and does not provide explicit action likelihoods, this formulation is not directly applicable.

Instead, we define a projection operator that is compatible with flow matching by introducing the following surrogate divergence:

$$D_{\text{FM}}(\pi^\star(\cdot \mid o), \pi_\theta(\cdot \mid o)) \; \triangleq \; \mathbb{E}_{a \sim \pi^\star(\cdot \mid o)}\big[\mathcal{L}_{\text{FM}}(\theta; o, a)\big], \tag{10}$$

which measures how well $\pi_\theta$ matches samples drawn from $\pi^\star$ under the flow-matching loss.

Using this divergence, the projection step becomes:

$$
\begin{aligned}
\theta^\star &= \arg\min_\theta \; \mathbb{E}_{o \sim \mathcal{D}} \, \mathbb{E}_{a \sim \pi^\star(\cdot \mid o)}\big[\mathcal{L}_{\text{FM}}(\theta; o, a)\big] \\
&\approx \arg\min_\theta \; \mathbb{E}_{(o,a) \sim \mathcal{D}}\Big[w(o, a) \, \mathcal{L}_{\text{FM}}(\theta; o, a)\Big],
\end{aligned}
\tag{11}
$$

where the approximation follows a standard offline RL practice that replaces sampling from $\pi^\star$ with weighted samples from a fixed dataset (Schulman et al., 2015). The weights are proportional to the exponential advantage: $w(o, a) \; \propto \; \exp\!\left(\frac{A^{\pi_{\text{ref}}}(o,a)}{\beta}\right)$.

Then, by setting the discount factor $\gamma \to 1$ and assigning a large negative reward to failure trajectories, Eq. 11 reduces to Eq. 4, which is the objective used in our policy update.

# B. Task Details

**Success Criteria.** We define task success using simple, outcome-based criteria that can be reliably judged from the final state (or a short post-action observation window):

- **Stacking:** Success if block $A$ is stably placed on top of block $B$ (with $A$ supported by $B$, not the table) and the stack remains upright for a short holding period.
- **Open Book:** Success if the front cover is opened beyond a predefined angle (e.g., clearly separated from the pages and lying open) and remains open at the end of the episode.
- **Erase Marks:** Success if all visible marker strokes are removed from the whiteboard area (i.e., no clearly detectable marks remain) at the end of the episode.
- **Scooping:** Success if at least a minimum amount of the target object $A$ is transferred into the bowl (with non-trivial contents remaining in the bowl at the end), while the majority of the transferred items are inside the bowl rather than spilled outside.

- **Drawing:** Success if the robot produces a single closed curve that forms a visually complete circle (i.e., endpoints meet with small gap tolerance) on the whiteboard within the designated drawing region.

**Detailed success rate improvement** All task is evaluated 50 times since we collect 50 online rollouts in each iteration. DSRL baseline is evaluated with 10 times since it's too time-consuming to evaluate too many rollouts during online update.

| Method | Stacking | Wiping | Open Book | Scooping | Drawing | Mean |
|---|---|---|---|---|---|---|
| Base model | 0.62 | 0.46 | 0.56 | 0.44 | 0.22 | 0.460 |
| DSRL | 0.70 | 0.40 | 0.50 | 0.60 | 0.30 | 0.500 |
| Filtered BC-1 | 0.80 | 0.62 | 0.72 | 0.64 | 0.46 | 0.648 |
| Filtered BC-2 | 0.88 | 0.76 | 0.82 | 0.74 | 0.56 | 0.752 |
| Ours-1 | 0.80 | 0.72 | 0.80 | 0.72 | 0.68 | 0.744 |
| Ours-2 | 0.92 | 0.86 | 0.86 | 0.92 | 0.78 | 0.868 |

*Table 2.* Detailed Success rates across 5 manipulation tasks.

## C. Reward Model Details

We use the Qwen3-VL-4B-Instruct model (Team, 2025a) as the vision–language reward model. Each trajectory is temporally downsampled into a 16-frame video before being fed to the model. We fintune the Qwen3-VL-4B-Instruct model for 200 steps with batch size 128.

We observe that directly prompting the reward model to output a binary `yes`/`no` decision can be overly optimistic, leading to a non-negligible number of false positives. To mitigate this issue, we instead examine the model-assigned probability of the ``yes'' token and only label a trajectory as successful when this probability exceeds a threshold of 0.8, with this threshold, model is more conservative on generate success label.

We compare this threshold-based criterion with the naive approach of directly querying the model for a binary answer. Empirically, using a higher confidence threshold substantially reduces the number of false-positive trajectories, resulting in more reliable supervision for downstream policy learning.

*Table 3.* Confusion matrices comparing the original reward model decision and our threshold-based criterion. We manually label a subset of 40 trajectories and compare the predictions of each method against human-annotated ground-truth labels. The false-positive number significantly dropped.

| **Original Method (Direct Yes/No Output)** | | |
|---|---|---|
| | **Predicted** | |
| | Success | Failure |
| **GT** Success | 15 | 7 |
| **GT** Failure | 8 | 10 |

| **Ours (Probability Threshold** $p(\text{yes}) > 0.8$**)** | | |
|---|---|---|
| | **Predicted** | |
| | Success | Failure |
| **GT** Success | 10 | 12 |
| **GT** Failure | 2 | 16 |

