# OpenReview forum: "VLAW: Iterative Co-Improvement of Vision-Language-Action Policy and World Model"
_ICML.cc/2026/Conference — ICML 2026 regular_

### Official Review · Reviewer_oELS · 2026-03-11

**Soundness:** 2
**Presentation:** 3
**Significance:** 3
**Originality:** 2
**Overall Recommendation:** 3
**Confidence:** 3

**Summary:**

This paper proposes an iterative framework that alternates between improving a world model using
real robot rollouts and refining a vision–language–action policy using filtered synthetic data
generated from the improved world model. By incorporating failure data to reduce overly optimistic
predictions for world model learning and using success-filtered rollouts for policy learning, the
method achieves consistent improvements across multiple contact-rich real-world manipulation
tasks.

**Compliance With Llm Reviewing Policy:**

Affirmed.

**Key Questions For Authors:**

1. The policy learning objective appears closely related to existing approaches such as
advantage-weighted regression and reward-filtered behavioral cloning. It would be helpful if
the authors could clarify what distinguishes the proposed method theoretically from these prior
formulations.
2. Since the practical update reduces to binary reward-filtered behavioral cloning without
advantage estimation or negative updates, in what sense is the regularized RL interpretation
essential?
3. The method is described as iterative co-improvement, yet experiments are limited to one or two
iterations. It would be helpful if the authors could comment on whether the joint updates are
expected to converge or stabilize, and how performance evolves as the number of iterations
increases.
4. The reward model provides only trajectory-level success or failure labels, without fine-grained
feedback on how the task is executed (e.g., speed, precision, smoothness, or efficiency), which
is often difficult to specify through language. Could this coarse supervision limit the model’s
ability to improve execution quality or scale to more nuanced behaviors as the VLA becomes
more capable?

**Limitations:**

yes

**Strengths And Weaknesses:**

Strengths:

1. The paper demonstrates consistent improvements across multiple contact-rich real-world
manipulation tasks, which strengthens the practical significance of the proposed approach.
2. The presentation is clear and well-organized. The iterative pipeline for jointly improving the
world model and policy is conceptually simple, easy to follow, and systematically designed,
making the overall approach practically scalable.

Weakness:

1. The theoretical contribution appears limited. The policy learning objective largely reframes
existing approaches such as advantage-weighted regression and reward-filtered behavioral
cloning, rather than introducing a fundamentally new policy optimization method. The primary
novelty lies in the overall algorithmic pipeline and system design.
2. The interpretation as a regularized RL procedure seems somewhat overstated. In practice, the
update reduces to filtered behavioral cloning with binary reward weighting, without advantage
estimation or negative updates from failure samples. This is closer to reward-filtered supervised
learning than to a full regularized RL formulation.
3. Although the algorithm is described as iterative co-improvement, the experiments only run for
two iterations, and no evidence is provided that the procedure converges or stabilizes. There is
no analysis of whether the joint policy and world model updates approach a fixed point or
whether the process is fundamentally heuristic. As such, it is unclear to what extent this
constitutes true co-improvement rather than a limited two-stage refinement.
4. Typo: rollous→rollouts (line 35),

---

> ### Author Rebuttal · Authors · 2026-03-31
>
> We sincerely appreciate your time and effort in reviewing our paper.
>
> ---
>
> **W1, W2, Q1: Theoretical contribution / the policy optimization method is not new**
>
> We agree that our policy update rule itself is not new. It can be viewed as advantage-weighted RL or reward-weighted behavioral cloning. We choose this class of methods because our base \(\pi_{0.5}\) policy is trained with flow matching, which does not provide a tractable policy log-likelihood for standard RL updates.
>
> Our main contribution is not a new policy optimization algorithm, but a practical world-model-based VLA improvement pipeline that combines three strong pretrained models.
>
> ---
>
> **W3, Q3: Why only two iterations**
>
> Each iteration in our pipeline is expensive: it requires real-world policy rollouts (about 1--2 hours), world model fine-tuning (about 8--12 hours), synthetic data generation, and policy fine-tuning (about 1--2 hours). Because of this cost, we only run two iterations in the paper.
>
> We also note that using only a small number of iterations is common for large-scale policy improvement. For example, \(\pi_{0.6}\) [1] also uses two rounds of iteration.
>
> [1] *\(\pi_{0.6}\): a VLA That Learns From Experience*. arXiv preprint arXiv:2511.14759
>
> ---
>
> **W4: Typo**
>
> Thank you for catching this. We will fix the typo in the final version.
>
> ---
>
> **Q2: The regularized RL interpretation is not necessary**
>
> We agree that this interpretation is not necessary for the practical algorithm. In practice, because our pipeline already involves three large pretrained models, we keep the training procedure as simple as possible. Instead of learning an additional advantage function, we use a VLM-based reward model to provide binary success labels and perform reward-filtered behavioral cloning.
>
> We include the regularized RL interpretation mainly to provide a more general perspective and connect our method to prior policy improvement formulations.
>
> ---
>
> **Q4: The reward model only provides trajectory-level success/failure labels, without fine-grained feedback**
>
> We agree. This is a limitation of the current method. Obtaining reliable fine-grained feedback for long-horizon manipulation is still difficult. We view this as an important direction for future work, especially as VLM-based reward models continue to improve.
>
> ---
>
> Thank you again for your constructive comments and support. We hope these responses address your concerns.

---

> > ### Author Rebuttal · Reviewer_oELS · 2026-04-05
> >
> > I appreciate the authors' responses.
> > They adequately addressed my concerns.

---

> > > ### Author Response · Authors · 2026-04-06
> > >
> > > Dear Reviewer oELS,
> > >
> > > Thank you for your thoughtful comments and we are glad that we were able to fully resolve your concerns. If you feel that our clarifications have resolved the issues you raised, we would greatly appreciate your reconsideration of the score.
> > >
> > > Please feel free to ask any further questions — we would be very happy to continue the discussion.
> > >
> > > Best regards,
> > > The Authors

---

### Official Review · Reviewer_SXV5 · 2026-03-12

**Soundness:** 3
**Presentation:** 3
**Significance:** 3
**Originality:** 3
**Overall Recommendation:** 4
**Confidence:** 4

**Summary:**

This paper proposes VLAW, a framework that iteratively improves vision-language-action (VLA) policies and video-based world models using limited real-world online interactions. It fine-tunes the world model with real rollouts to boost physical fidelity. The updated world model then generates synthetic trajectories, which are filtered by a fine-tuned VLM reward model, to further train the VLA policy. Overall, the main issue examined by the article is the bottleneck of costly real-world data collection for policy optimization. The authors strive to examine the aspect of utilizing a learned action-conditioned simulator to generate high-fidelity physical data for scalable policy training.

**Compliance With Llm Reviewing Policy:**

Affirmed.

**Final Justification:**

The author's response addressed my concerns. I will raise my score.

**Key Questions For Authors:**

What are the specific computational costs and time overheads for generating large-scale synthetic trajectories (e.g., 500) and fine-tuning the video diffusion world model during each iteration?

What is the false-positive rate (identifying failures as successes) of the VLM reward model in the current experiments? How can reward reliability be ensured as tasks become more fine-grained?

Can you provide performance comparisons against stronger baselines, such as standard image/video data augmentation techniques or other offline RL algorithms?

**Limitations:**

The limitations discussion could be strengthened. Beyond the limited task variety, I recommend exploring the risk of error accumulation (such as hallucinations) from using a VLM as a reward model, as well as the computational overhead of iteratively fine-tuning large video diffusion models.

**Strengths And Weaknesses:**

Strengths:

Originality & Significance: Correcting the world model's over-optimistic bias using real-world failure data to create a closed-loop policy training pipeline is a practical and sound approach.

Soundness: Evaluated on the real-world DROID robot platform, successfully tackling challenging contact-rich and deformable-object tasks.

Weaknesses:

Soundness (Evaluation): Evaluation is limited to only 5 task categories. For a framework targeting scalable, generalist policies, this is insufficient to prove broad generalization.

Soundness (Baselines): The baselines are weak (only Filtered BC and DSRL). It lacks comparisons with other visual data augmentation techniques or mainstream model-based RL methods.

Significance (Scalability): Relying on a simple VLM probability threshold for reward filtering is fragile. In longer-horizon or more complex tasks, VLM hallucinations could severely bottleneck system performance.

---

> ### Author Rebuttal · Authors · 2026-03-31
>
> We sincerely appreciate your time and effort in reviewing our paper.
>
> ---
>
> **W1 (Evaluation):**
> We agree that the evaluation of more tasks is important. However, real-world experiments are time-consuming: each task requires hundreds of robot rollouts across data collection, adaptation, and evaluation, which already takes more than 5 hours per task. To strengthen the evidence within the rebuttal period, we additionally provide results in the DROID simulation below.
>
> ---
>
> **W2 (Baseline):**
> Thank you for this suggestion. We added a model-based guided visual foresight(GVF) baseline [2] in the Polaris benchmark [1], which contains multiple simulated DROID tasks. For all methods, we use the same base policy, the same world model, and 50 online rollouts per task. Due to the rebuttal time limit, we conduct one round of improvement. VLAW consistently outperforms both the base policy and GVF:
>
> | Polaris Tasks      | FoodBussing | PanClean | MoveLatteCup | TapeIntoContainer |
> |--------------------|-------------|----------|--------------|-------------------|
> | Base policy        | 0.16        | 0.40     | 0.44         | 0.36              |
> | Guided Visual Foresight [2]            | 0.20        | 0.42     | 0.52         | 0.44              |
> | VLAW (ours)        | **0.34**    | **0.56** | **0.64**     | **0.52**          |
>
> *Base policy: official \(\pi_{0.5}\) checkpoint in Polaris: \texttt{gs://openpi-assets/checkpoints/polaris/pi05\_droid\_jointpos\_polaris}.*
>
> [1] *PolaRiS: Scalable Real-to-Sim Evaluations for Generalist Robot Policies*
> https://arxiv.org/abs/2512.16881
>
> [2] *Improvisation through Physical Understanding: Using Novel Objects as Tools with Visual Foresight*
> https://arxiv.org/abs/1904.05538
>
> ---
>
> **W3 (Scalability / VLM accuracy):**
> We agree that VLM accuracy is important. To improve reliability, we fine-tune the VLM on our tasks and use a confidence threshold to reduce false positives.
>
> ---
>
> **Q1 (Computational cost):**
> On a single A100 GPU, generating one synthetic trajectory takes about 100 seconds. World model fine-tuning for 30k steps takes about 12 hours. Synthetic rollout generation can be parallelized across GPUs.
>
> ---
>
> **Q2 (False-positive rate):**
> We manually checked a subset of trajectories and report the confusion matrix in Appendix C:
>
> |                  | Predicted Success | Predicted Failure |
> |------------------|------------------|-------------------|
> | GT Success       | 10               | 12                |
> | GT Failure       | 2                | 16                |
>
>
> ---
>
> **Q3 (Stronger baseline):**
> We added the model-based guided visual foresight (GVF) baseline as discussed above. Standard RL baselines are difficult to apply to \(\pi_{0.5}\) because it is trained with flow matching and does not provide a tractable policy log-likelihood for RL updates.
>
> ---
>
> Thank you again for your constructive comments and support. We hope these responses address your concerns.

---

> > ### Author Rebuttal · Reviewer_SXV5 · 2026-04-03
> >
> > Thank you for the additional experiments. The Polaris simulation results and computational cost details are helpful.
> >
> > However, my main concerns are only partially addressed: the added baseline (GVF, 2019) is not strong enough, and the VLM reward scalability concern was not substantively addressed with new analysis.
> > I maintain my current score.

---

> > > ### Author Response · Authors · 2026-04-04
> > >
> > > Dear Reviewer SXV5,
> > >
> > > Thank you for your feedback on our rebuttal and for your additional questions. We would like to provide the following clarifications:
> > >
> > > - **Why we chose GVF as the baseline instead of other baselines**
> > >   1. Many standard RL algorithms are **very difficult to apply to the state-of-the-art π0.5 VLA**, since π0.5 is trained with flow-matching its policy log-likelihood is intractable. As a result, a large class of likelihood-based RL methods cannot be straightforwardly applied in our setting. We therefore chose GVF, DSRL, and Filtered BC as baselines because they **do not** explicitly require policy log-likelihood, and all represent **typical paradigms** for RL-based policy improvement.
> > >   2. More broadly, **scaling prior model-based RL** methods to large pretrained models remains **highly challenging** in practice. Our framework already involves training and coordinating three large pretrained models, and we have already invested **substantial effort** in implementing these baselines in our setting.
> > >
> > > - **Regarding the scalability of VLM-based rewards**
> > >   1. Training a VLM that can serve as a general-purpose reward model is itself a challenging research problem and **is worthy of dedicated study** [1]. In this work, however, our focus is on **world-model-based VLA policy improvement** rather than reward-model pretraining. Therefore, we adopt a practical design choice and fine-tune a strong open-source model, Qwen3-VL, as our reward model.
> > >   2. We also believe this direction will naturally benefit from continued advances in VLM capabilities. As stronger off-the-shelf VLMs become available, the reward model component in our framework can be improved accordingly.
> > >
> > > [1] *RoboReward: General-Purpose Vision-Language Reward Models for Robotics* https://arxiv.org/pdf/2601.00675
> > >
> > > We hope these clarifications address your concerns. We sincerely appreciate your thoughtful questions and would be happy to provide further clarification if helpful.
> > >
> > > Best regards,
> > > Authors

---

### Official Review · Reviewer_vUdT · 2026-03-12

**Soundness:** 2
**Presentation:** 3
**Significance:** 3
**Originality:** 2
**Overall Recommendation:** 3
**Confidence:** 3

**Summary:**

The paper proposes VLAW, an iterative framework that improves a pretrained VLA policy by first grounding a pretrained action-conditioned world model using limited real-world online rollouts, then using the improved world model to generate large-scale synthetic trajectories, and finally updating the VLA with successful real and synthetic trajectories selected by a vision-language reward model. Experiments on five real-robot contact-rich/deformable-object task categories on DROID show that the grounded world model predicts interaction outcomes better than the pretrained or expert-only-finetuned baselines. Also, the full method improves the mean success rate from 46% for the base model to 86%.

**Compliance With Llm Reviewing Policy:**

Affirmed.

**Final Justification:**

I appreciate the authors' rebuttal and the clarifications they provided. I agree that making three large pre-trained models work together in a stable real-world policy improvement framework requires substantial engineering effort and has practical value.

However, my final assessment remains weak reject. My main concern is that, although the paper may be the first to combine a world model with a flow-matching VLA policy in this real-world setting, the methodological novelty still seems limited. As presented, the contribution feels stronger as a systems/robotics integration effort than as a clear ML-method advance at the level I would expect for ICML.

I also remain unconvinced on soundness, mainly because the experimental evaluation is relatively limited in terms of task diversity, baselines, and evidence of broader generalization. The rebuttal helped explain why GRPO-style baselines may not be applicable to $\pi$-0.5 due to the lack of tractable policy likelihood, and I appreciate that clarification. However, this only partially addresses the concern, since the empirical comparison is still somewhat narrow overall.

Overall, I see moderate practical significance and reasonable real-world value, but only moderate originality and insufficient empirical support for stronger confidence. The rebuttal improved clarity but did not substantially change my evaluation, so I will keep my recommendation as a weak reject.

**Key Questions For Authors:**

- How much of the improvement comes from the proposed iterative design itself, versus simply starting from strong pretrained components?
- The paper argues that existing world models lack sufficient physical fidelity for policy improvement. Can the authors provide a direct baseline that uses such a pretrained world model for policy improvement without the proposed adaptation?
- Does the iterative procedure continue to improve beyond two rounds, or does it begin to saturate or drift?

**Limitations:**

yes

**Strengths And Weaknesses:**

## Strengths
- The paper tackles an important bottleneck in VLA improvement: real-world rollout collection is costly, especially for contact-rich and deformable-object tasks.
- The central motivation is clear: pretrained world models are often not physically reliable enough for policy improvement without additional grounding.
- The main policy comparisons use matched real-rollout budgets, which makes the benefit of synthetic rollouts more credible.
- The paper is easy to follow at a high level; the problem setup, method, and takeaways are communicated clearly.

## Weaknesses
- The baseline set is narrow. There is no direct comparison to another world-model-based VLA improvement method in the same real-robot setting.
- It is unclear how much of the gain comes from the proposed training procedure versus the use of strong pretrained components.
- The method relies on multiple large pretrained models, including a VLA, a world model, and a reward model, which may limit reproducibility and accessibility.
- The Generalization ability for under lower-coverage online data or larger distribution shifts remains unclear.
- The overall method is a strong integration of existing ingredients, so the novelty claim would benefit from sharper empirical differentiation from related concurrent work.

---

> ### Author Rebuttal · Authors · 2026-03-31
>
> We sincerely appreciate your time and effort in reviewing our paper.
>
> ---
>
> **W1: About the baseline.**
>
> Thank you for this valuable suggestion. To make the comparison more rigorous and reproducible, we evaluated our method in the Polaris benchmark [1], which includes multiple simulated DROID tasks. We also added a model-based visual foresight baseline, GVF [2], which learns a world model and selects the action with the best predicted outcome.
>
> For all methods, we collect 50 online rollouts for each task. GVF uses the same world model as the planner, ensuring a fair comparison. Due to the rebuttal time constraint, we conduct one round of policy improvement. VLAW consistently outperforms the prior method:
>
> | Polaris Tasks      | FoodBussing | PanClean | MoveLatteCup | TapeIntoContainer |
> |--------------------|-------------|----------|--------------|-------------------|
> | Base policy        | 0.16        | 0.40     | 0.44         | 0.36              |
> | Guided Visual Foresight [2]            | 0.20        | 0.42     | 0.52         | 0.44              |
> | VLAW (ours)        | **0.34**    | **0.56** | **0.64**     | **0.52**          |
>
> *The base policy is the official \(\pi_{0.5}\) checkpoint in Polaris: (gs://openpi-assets/checkpoints/polaris/pi05\_droid\_jointpos\_polaris)
>
> [1] PolaRiS: Scalable Real-to-Sim Evaluations for Generalist Robot Policies*
> https://arxiv.org/abs/2512.16881
>
> [2] Improvisation through Physical Understanding: Using Novel Objects as Tools with Visual Foresight*
> https://arxiv.org/abs/1904.05538
>
> ---
>
> **W2, W5, Q1: About novelty and how much of the gain comes from the proposed training procedure.**
>
> Our goal is to develop a **scalable and effective** pipeline for multi-task policy improvement. To this end, we build on three pretrained models in our framework.
>
> We think using three pretrained models to perform model-based VLA improvement is novel. And since the models are huge, we keep the training process as simple as possible
>
> ---
>
> **W3: The method relies on multiple large pretrained models, which may limit reproducibility and accessibility.**
>
> We agree that reproducibility is important. Importantly, all three pretrained models used in our framework (Qwen3-VL, \(\pi\), and Ctrl-World) are fully open-source. To further improve reproducibility, our training pipeline uses weighted supervised fine-tuning, which is substantially simpler and more stable than many reinforcement-learning-based alternatives. We will also release the full training code upon acceptance.
>
> ---
>
> **W4: The generalization ability under lower-coverage online data or larger distribution shifts remains unclear.**
>
> We agree with this concern. Our current results suggest that the fine-tuned world model mainly improves prediction quality around the policy rollout distribution seen during online data collection.
>
> It does not yet generalize reliably to entirely new tasks or to large distribution shifts. We view this as an important direction for future work.
>
> ---
>
> **Q2: Direct baseline using the pretrained world model without adaptation on online rollouts.**
>
> We include examples of synthetic data generated by the pretrained world model before online adaptation in Figure 6. These predictions are noticeably blurrier, which can harm the VLA policy.
>
> ---
>
> Thank you again for your constructive comments and support. We hope these responses address your concerns, and we would be happy to clarify any additional questions.

---

> > ### Author Rebuttal · Reviewer_vUdT · 2026-04-05
> >
> > Thank you for the rebuttal and the additional Polaris results. I appreciate the extra experiments. However, my main concerns remain.
> >
> > I still think the novelty claim would benefit from further clarification. In particular, combining three pretrained models does not yet make it fully clear how much of the improvement comes from the proposed training procedure itself, as opposed to the pretrained components.
> >
> > The added GVF baseline is valuable, and I believe the paper would be further strengthened by including a more direct comparison to another world-model-based VLA method in a similar setting.
> >
> > Overall, the rebuttal improves the paper, but it does not change my assessment.

---

> > > ### Author Response · Authors · 2026-04-06
> > >
> > > Dear Reviewer vUdT,
> > >
> > > Thank you again for your thoughtful feedback. We would like to briefly clarify the following points.
> > >
> > > **1. On novelty.**
> > > Our key contribution is a **real-world, world-model-based policy improvement framework for a state-of-the-art flow-matching π0.5  policy**. To the best of our knowledge, this is the first work to improve a flow-matching VLA policy with a world model in the real world, including contact-rich and deformable-object tasks.
> > >
> > > Since we already scale method with three large pre-trained models, we keep the training procedure **as simple as possible to make it stable**. As a result, the procedure itself is not entirely new.  However, **as Reviewer ChBb noted** in Rebuttal Acknowledgement, making it work with three large pre-trained models **requires solid system effort and have real-world value** (https://openreview.net/forum?id=68zOaa2gOf&noteId=jYiregfVwM).
> > >
> > > ---
> > >
> > > **2. On baselines.**
> > > We do not compare with recent GRPO-style world-model-based VLA methods (e.g., concurrent works such as WMPO [1] and World-Gymnast [2]) because they are **not applicable** to π0.5. Since π0.5 is trained with a flow-matching objective, it **does not provide a tractable policy log-likelihood**, which GRPO-style optimization requires.
> > >
> > > We therefore compare with baselines that do not require explicit policy likelihood, including **DSRL, Filtered BC, and GVF**.
> > >
> > > ---
> > >
> > > We hope these clarifications help address your concerns. We sincerely appreciate your thoughtful questions and would be happy to provide further clarification if needed.
> > >
> > > Best regards,
> > > The Authors
> > >
> > > ---
> > >
> > > [1] WMPO: World Model-based Policy Optimization for Vision-Language-Action Models https://arxiv.org/abs/2511.09515
> > >
> > > [2] World-Gymnast: Training Robots with Reinforcement Learning in a World Model https://arxiv.org/abs/2602.02454

---

### Official Review · Reviewer_ChBb · 2026-03-15

**Soundness:** 2
**Presentation:** 3
**Significance:** 3
**Originality:** 3
**Overall Recommendation:** 4
**Confidence:** 4

**Summary:**

The paper proposes VLAW, a framework that iteratively improves a vision-language-action (VLA) policy and a video-based world model, using real-world rollouts to improve the world model and then using the improved world model to generate synthetic data to further improve the policy.

**Compliance With Llm Reviewing Policy:**

Affirmed.

**Final Justification:**

Increased from 3 to 4: while the rebuttal does not fully convince me that the method is fundamentally more than well-engineered data filtering and augmentation, I feel that this is a solid systems-oriented paper with some real-world value.

**Key Questions For Authors:**

- To assess the strength of the synthetic rollouts, how much real-world data would be needed to achieve similar improvements without synthetic rollouts?

- To assess the effectiveness of the reward model, what is the error rate of the reward model on unseen trajectories?

- To assess world model stability, how does prediction error accumulate during long-horizon imagined rollouts?

- Does the method generalize to tasks significantly different from those used to fine-tune the world model?

**Limitations:**

Yes

**Strengths And Weaknesses:**

Strengths:
- The approach pursues a solution to a real bottleneck: real-world data collection for robot learning

- The manuscript is well-written and clear

- The method displays strong engineering (in the combination of VLA policies, video world models, and vision-language reward models) and empirical contributions (in the reported performance gains).

Weaknesses:

- L97-103,C2 + Section 5: I find the baseline comparisons to be somewhat underwhelming. The manuscript’s main contribution is in the iterative co-improvement of the world model and the policy (as opposed to increased model capacity and data scale) — to validate the need for the world model. Therefore, comparison with some of these “earlier methods”, but with comparable capacity and scale to the proposed approach, would be more instructive.

- L202-209,C2: The policy is trained only on successful trajectories generated by the world model, which may itself introduce a strong bias toward optimistic trajectories. It is unclear whether this improves robustness or merely reinforces existing behavior.

- L204-212,C1: The reward model labels success based on a vision-language model with a probability threshold; it could also serve as a single point of failure, should it mislabel synthetic trajectories or introduce noise into policy training.

- L368-377, Figure 5: Long-horizon world model rollouts are known to accumulate error. The paper shows examples up to 20 seconds, but systematic evaluation of rollout error accumulation is limited.

---

> ### Author Rebuttal · Authors · 2026-03-31
>
> We sincerely appreciate your time and effort in reviewing our paper！
>
> ---
>
> **W1: More comparison with previous methods at the same scale.**
>
> Thank you for this valuable suggestion. To make the comparison more rigorous and reproducible, we evaluated our method in the Polaris benchmark [1], which includes multiple simulated DROID tasks. We also added a model-based visual foresight baseline, Guided Visual Foresight (GVF) [2], which learns a world model and selects the action with the best predicted outcome.
>
> For all methods, we collect 50 online rollouts per task. GVF uses the same world model as the planner, ensuring a fair comparison. Due to time constraints on rebuttals, we report only one round of improvement. VLAW consistently outperforms both the base policy and GVF:
>
> | Polaris Tasks      | FoodBussing | PanClean | MoveLatteCup | TapeIntoContainer |
> |--------------------|-------------|----------|--------------|-------------------|
> | Base policy        | 0.16        | 0.40     | 0.44         | 0.36              |
> | Guided Visual Foresight [2] | 0.20 | 0.42 | 0.52 | 0.44 |
> | VLAW (ours)        | **0.34**    | **0.56** | **0.64**     | **0.52**          |
>
> *The base policy is the official \(\pi_{0.5}\) checkpoint in Polaris: \texttt{gs://openpi-assets/checkpoints/polaris/pi05\_droid\_jointpos\_polaris}.*
>
> [1] *PolaRiS: Scalable Real-to-Sim Evaluations for Generalist Robot Policies*
> https://arxiv.org/abs/2512.16881
>
> [2] *Improvisation through Physical Understanding: Using Novel Objects as Tools with Visual Foresight*
> https://arxiv.org/abs/1904.05538
>
> ---
>
> **W2: The policy is trained only on success data. Does the method improve robustness, or merely reinforce existing behavior?**
>
> Thank you for this insightful question. Even for the same input, the base \(\pi_{0.5}\) policy still produces stochastic outputs. Our method uses the world model to search among these possible behaviors and identify trajectories with better predicted outcomes. Training on these filtered trajectories shifts the policy toward more successful behaviors, rather than simply repeating its original distribution.
>
> ---
>
> **W3 and Q2: Reward model error rate and its effect on the policy.**
>
> Yes, the reward model can still produce false positives even after fine-tuning, as shown in Appendix C. However, the policy can still improve because false positives are relatively rare, while most retained synthetic trajectories are true successes.
>
> |                  | Predicted Success | Predicted Failure |
> |------------------|------------------|-------------------|
> | GT Success       | 10               | 12                |
> | GT Failure       | 2 (false positive) | 16              |
>
> (Table in Appendix C.)
>
> ---
>
> **W4 and Q3: Systematic evaluation of rollout error accumulation.**
>
> We agree this is important. After fine-tuning on online policy rollouts, the world model can generate rollouts up to 20 seconds. Following your suggestion, we evaluated video quality under different rollout horizons:
>
> | Rollout Horizon | 5s | 10s | 20s |
> |-----------------|----|-----|-----|
> | PSNR \(\uparrow\) | **23.32** | 22.63 | 21.77 |
> | FVD \(\downarrow\) | **57.22** | 60.35 | 64.12 |
>
> As expected, quality degrades with longer rollout horizons due to error accumulation, but the model remains reasonably stable even at 20 seconds.
>
> ---
>
> **Q4: How much real-world data is needed to achieve similar improvements without synthetic rollouts?**
>
> This is an important question. Since real-world rollouts are expensive, it is difficult to determine the exact amount needed to match the gain from synthetic data. As a rough comparison, Figure 7 shows that filtered BC using only real-world data after two iterations (100 rollouts) achieves a similar average success rate to VLAW after one iteration (50 rollouts). This suggests that synthetic rollouts can substantially improve data efficiency.
>
> ---
>
> Thank you again for your constructive comments and support. We hope these responses address your concerns, and we would be happy to clarify any additional questions.

---

> > ### Author Rebuttal · Reviewer_ChBb · 2026-04-05
> >
> > I appreciate the authors' responses.
> >
> > While the rebuttal does not fully convince me that the method is fundamentally more than well-engineered data filtering and augmentation, I feel that this is a solid systems-oriented paper with some real-world value.
> >
> > Increasing from 3 to 4.

---

> > > ### Author Response · Authors · 2026-04-06
> > >
> > > Dear Reviewer ChBb,
> > >
> > > We are very glad that we were able to fully resolve your concerns. Making VLAW work effectively with three pre-trained models required substantial systems effort, and we sincerely appreciate your recognition of this aspect of our work.
> > >
> > > Best regards,
> > > The Authors

---

### Decision · Program_Chairs · 2026-04-30

**Decision:**

Accept (regular)

**Comment:**

This paper looks at how to improve performance of a vision language action model (VLA) with iterative real-world rollouts using an action conditioned video world model.  The problem here is important -- real world data collection and rollouts are a serious challenge, and goes about it with solid engineering and system design.

The core idea -- using the real-world rollouts to improve a world model and use that for training -- is useful, and the experimental design is solid. The reviewers raised concerns about novelty (it is a combination of existing large models), baselines, and only testing on a handful of tasks -- but given the difficulty of real-world experiments and of running RL on a large VLA like pi-0.5, these weaknesses seem forgiveable.

It seems like a worthwhile contribution to the problem of efficiently improving large robotics models in the real world.